# Pharmacological Inhibition of CA-IX Impairs Tumor Cell Proliferation, Migration and Invasiveness

**DOI:** 10.3390/ijms21082983

**Published:** 2020-04-23

**Authors:** Valerio Ciccone, Arianna Filippelli, Andrea Angeli, Claudiu T. Supuran, Lucia Morbidelli

**Affiliations:** 1Department of Life Sciences, University of Siena, 53100 Siena, Italy; ciccone3@student.unisi.it (V.C.); arianna.filippelli@student.unisi.it (A.F.); 2Dipartimento NEUROFARBA, Sezione di Scienze Farmaceutiche, Università di Firenze, Sesto Fiorentino, 50139 Firenze, Italy; andrea.angeli@unifi.it (A.A.); claudiu.supuran@unifi.it (C.T.S.)

**Keywords:** Carbonic anhydrase IX, cancer cells, apoptosis, epithelial-mesenchymal transition, invasiveness

## Abstract

Carbonic anhydrase IX (CA-IX) plays a pivotal role in regulation of pH in tumor milieu catalyzing carbonic acid formation by hydrating CO_2_. An acidification of tumor microenvironment contributes to tumor progression via multiple processes, including reduced cell-cell adhesion, increased migration and matrix invasion. We aimed to assess whether the pharmacological inhibition of CA-IX could impair tumor cell proliferation and invasion. Tumor epithelial cells from breast (MDA-MB-231) and lung (A549) cancer were used to evaluate the cytotoxic effect of sulfonamide CA-IX inhibitors. Two CA-IX enzyme blockers were tested, SLC-0111 (at present in phase Ib clinical trial) and AA-06-05. In these cells, the drugs inhibited cell proliferation, migration and invasion through shifting of the mesenchymal phenotype toward an epithelial one and by impairing matrix metalloprotease-2 (MMP-2) activity. The antitumor activity was elicited via apoptosis pathway activation. An upregulation of p53 was observed, which in turn regulated the activation of caspase-3. Inhibition of proteolytic activity was accompanied by upregulation of the endogenous tissue inhibitor TIMP-2. Collectively, these data confirm the potential use of CA-IX inhibitors, and in particular SLC-0111 and AA-06-05, as agents to be further developed, alone or in combination with other conventional anticancer drugs.

## 1. Introduction

Development of many solid tumors implies the activation of aberrant growth and survival signals that drastically reprogram cancer cell energy metabolism [1]. Deregulated energy metabolism and inadequate perfusion cause the remodeling of tumor microenvironment, including hypoxia, pH balance, changing in glucose and lactate secretion and recruitment of immune and stromal cells [2]. To maintain an intracellular pH (pHi) compatible with cell survival and proliferation, a large family of transporters extrudes the excess of lactate and protons in the extracellular milieu. Subsequently, the pHi is increased compared to normal cells, while the extracellular pH (pHe) is decreased. This “reversed pH gradient” in cancer cells is considered an emerging hallmark of cancer [3], promoting tumor growth and metastasis via various mechanisms. Among them, there are pH-dependent degradation of the extracellular matrix via activation of metalloproteases and cathepsins, and modulation of integrin-mediated cell matrix adhesion [4]. Furthermore, alkaline pHi can foster cancer cell migration by cytoskeletal reorganization [5]. Finally, an acidification of pHe can influence the invasive and metastatic potential of tumors by epithelial-mesenchymal transition (EMT) induction [6,7].

Several proteins are involved in the pH regulation including monocarboxylate transporters 1 and 4 (MCT1 and MCT4), Na^+^/H^+^ exchangers (NHE1), Na^+^/HCO_3_^-^ co-transporters and carbonic anhydrase IX (CA-IX) [4].

CA-IX belongs to the CA family, ubiquitous zinc enzymes, encoded by eight distinct, evolutionary unrelated gene families. The biochemical activity of CAs are the reversible hydration of CO_2_ to HCO_3_^−^ and H^+^ playing essential role in several cellular processes including pHe and pHi regulation, respiration, metabolism and other processes that require CO_2_and HCO_3_^−^ [8].

Since the membrane localization of CA-IX, this enzyme plays a pivotal role in the regulation of pHi and pHe. Indeed, CA-IX is normally expressed in gut epithelial cells [9], but it is upregulated in many solid tumors including breast and lung [10,11,12,13]. Its expression is under control of the hypoxia inducible factor-1α (HIF-1α) and it is localized in chronically hypoxic tumor regions. Accumulating translational evidences suggest that CA-IX expression correlates with poor prognosis [14], because it is induced by hypoxia, functionally linked to acidosis, tumor invasiveness and drug resistance [15].

The aim of this work was to determine whether CA-IX could represent a novel target for breast and lung tumor treatment. Here, we showed that pharmacological inhibition of CA-IX induces tumor cell death, with activation of the apoptosis cascade, and reduction of invasiveness through shifting of the mesenchymal phenotype toward an epithelial one and impairment of gelatinolytic activity.

We took advantage of the availability of novel CA IX inhibitors, SLC-0111 and AA-06-05. SLC-0111 is a ureido benzene-sulfonamide, currently in phase Ib clinical trial for the treatment of solid tumors associated to hypoxic micro-environments [16,17,18]. While SLC-0111 has high affinity for CA-IX, AA-06-05 is an inhibitor also of CA-I and CA-II isoforms. CA-I and CA-II are cytoplasmatic isoforms, both involved in several biological pathway, such as respiration, the acid/base homeostatic regulation, due the reversible hydration of CO_2_, and bone resorption [19].

## 2. Results

### 2.1. Characterization of CA-I, CA-II and CA-IX Expression in Tumor Cells

We examined the basal expression of CA-I, CA-II and CA-IX in a panel of cell lines from different cancer tissues: MDA-MB-231, metastatic breast cancer cells, and A549, non-small cell lung cancer cells. Using western blot analysis, we could detect CA-IX protein in both cell lines, with higher intensity and multiple bands in MDA-MB-231 respect to A549, where only one band with lower intensity was visible (Figure 1A,B). In the same samples, we found CA-I expression only in MDA-MB-231 (Figure 1A,B), while CA-II was not detected in either cell lines (data not shown).

As CA-IX is ectopically expressed in tumors, but it is one of the most upregulated gene in a HIF-1α dependent manner [13,20], we assessed the regulation of CA-IX expression in hypoxic condition. Using CoCl_2_ to mimic hypoxia condition, we did not observe an increase of CA-IX expression in both cell lines (Figure 1C,D). On the bases of these results, we performed all the experiments in normoxia conditions.

### 2.2. CA-IX Pharmacological Inhibition Induces Cell Death in Tumor Cells

To test whether the inhibition of CA-IX with AA-06-05 and SLC-0111 could reduce cancer cell survival, the colorimetric MTT assay was performed on MDA-MB-231 (Figure 2A,B) and A549 (Figure 2C,D). The assay was performed in medium supplemented with 1% FBS, evaluating the effect of increasing concentrations of the CA-IX inhibitors [10–300 µM] after 48 h of treatment (Figure 2). Medium with 0.1% FBS was used as negative control of scarce growth. An evident concentration-dependent inhibitory effect was observed with high doses, ranging from 100 µM to 300 µM, of both CA-IX inhibitors. In particular, treatment with AA-06-05 [100–300 µM] had a stronger effect on cancer cell viability, especially on MDA-MB-231 cells (Figure 2 B,D).

These data indicate that pharmacological targeting the CA-IX in tumor cells produces an impairment of cell survival.

### 2.3. CA-IX Pharmacological Inhibition Activates Apoptotic Pathway in Tumor Cells

From a molecular point of view, we focused on assessing whether the inhibition of CA-IX determines a modulation of apoptotic pathways. Therefore, to evaluate the effect of pharmacological CA-IX inhibitors on MDA-MB-231 and A549 cells, the expression of apoptotic proteins was evaluated by western blot. To quantify the principal apoptotic biomarkers, and their activation, in response of increasing concentrations of SLC-0111 and AA-06-05, cells were exposed to concentrations of 100–200 µM of each pharmacological inhibitor. 

Considering the role of CA-IX in the regulation of tumor cell metabolism and regulation of cellular pH and reactive oxygen species (ROS) accumulation [21], the activation of ERK1/2, a signaling molecule involved in both proliferation and oxidative stress-induced apoptosis, was assessed. The increase of p-ERK1/2 was evaluated in relation to total ERK1/2. The level of activated p-ERK1/2 arose after 30 min of incubation of MDA-MB-231 with both CA-IX inhibitors: AA-06-05 [100–200 µM] and SLC-0111 [100–200 µM] increased the expression of phosphorylated ERK1/2, compared to the basal control (growth condition of 1% FBS) and vehicle (Figure 3).

Based on the ability of p-ERK1/2 to activate p53 [22,23,24], we investigated the expression of this key protein in the apoptotic cascade. In our experiments p53 maximum level was observed after 60 min of incubation with the CA-IX pharmacological inhibitors (Figure 4). Data showed that the expression of p53 increased with 200 µM AA-06-05 in MDA-MB-231 (Figure 4A,B) and with 100 and 200 µM SLC-0111 and AA-06-05, compared to controls inA549 (Figure 4C,D).

It is reported that ERK1/2 signaling converges on p53-dependent Caspase-3 cleavage pathway [23,25]. The pro-apoptotic effect of CA-IX inhibitors was observed through the expression of Caspase 3, a key factor downstream of this pathway and, finally, its activated form, the cleaved-Caspase 3 (cl-Caspase 3). MDA-MB-231 showed an activation of caspase-3 mediated apoptosis after treatment with both inhibitors SLC-0111 [100–200 µM] and AA-06-05 [200 µM], strengthening a greater sensitivity of this cell type to CA-IX inhibition (Figure 5A,B). Caspase-3 activation was visible in A549 mainly after treatment with AA-06-05 [100–200 µM] (Figure 5C,D). 

All together, these data suggest that the inhibition of CA-IX, by SLC-0111 and AA-06-05, increases the activation of apoptotic pathways on tumor cells by the activation of p-ERK1/2 - p53 pathway, corroborating the functional results of survival assay.

### 2.4. Pharmacological Inhibition of CA-IX Reduces Tumor Cell Migration and Invasion Ability

In cancer progression, the metastatic process involves cell invasiveness [26]. Next, we analyzed the ability of CA-IX inhibitors to interfere with invasion of cancer cells by using the Boyden chamber assay. This assay was performed by exposing tumor cell suspensions (treated with 50 µM of CA-IX inhibitors, a subtoxic concentration) toward gradients of serum obtained by putting medium with 1 and 5% FBS in the lower compartments. In basal condition, the gradient induced by 5% FBS was responsible for the higher migrating cell number, especially with MDA-MB-231 compared to A549 (Figure 6). A significant reduction of cell migration caused by CA-IX pharmacological inhibition was observed in both tumor cell lines (Figure 6). AA-06-05 showed a more evident effect on tumor cell invasion ability compared to SLC-0111 (Figure 6B,D). A stronger inhibitory effect was observed in MDA-MB-231 (Figure 6A,B) in both chemotactic conditions (medium with 1% FBS and medium at 5% FBS in the lower compartment), being the inhibitory effect more significant with 5% FBS. 

In the complex, CA-IX inhibition exerts an inhibitory effect on tumor cell migration. The higher efficacy on MDA-MB-231 compared to A549 may be due to their mesenchymal-like phenotype, characterized by higher invasive potential.

Migration and invasion are both key components of EMT [27]. To corroborate the inhibitory effect on tumor cell migration (Figure 6), the impact of CA-IX inhibition was investigated on the EMT process. In agreement with literature, in basal condition, we found a greater expression of the epithelial marker, E-Cadherin, in A549 compared to MDA-MB-231 [28] (Figure 7A,B). MDA-MB-231 cells treated with SLC-0111 showed a slight decrease of mesenchymal marker Vimentin and increase of E-Cadherin, the epithelial marker (Figure 7C,D). In the same cells, a greater effect has been observed when treated with AA-06-05, corroborating the higher antitumor activity of this compound (Figure 7C,D). Overlapping results on EMT markers were obtained when A549 were exposed to CA-IX inhibitors (Figure 7E,F), with higher effects induced by AA-06-05. The mesenchymal marker Fibronectin was not significantly modulated by CA-IX inhibition (Figure 7).

These results suggest an impairment of EMT process in tumor cells treated with CA-IX inhibitors, confining the cells in an epithelial stage.

### 2.5. Inhibition of CA-IX is Related to Inhibition of MMP-2 Activity

Matrix metalloproteinases (MMPs) and their endogenous inhibitors (TIMPs) regulate EMT and are critical for cancer cell invasion and extracellular matrix degradation [29]. Finally, to evaluate the activity of MMP-2 in cells treated with CA-IX inhibitors, we performed gelatin zymographic analysis. The activation of released MMPs allows the degradation of gelatin contained in polyacrylamide gels. An evident in Figure 8A,D, gelatinase activity is observed under basal condition at 68–72 kDa, in MDA-MB-231 and A549, thus corroborating the activation of MMP-2. Consistently with migration and EMT related results, SLC-0111 and AA-06-05 showed a decreased MMP-2 enzymatic activity in both cells lines, being MDA-MB-241 more responsive (Figure 8A,B). In both cell lines the compound AA-06-05 was the most effective (Figure 8). ReducedMMP-2 gelatinolytic activity was accompanied by increased expression and release of TIMP-2, an endogenous inhibitor of MMP-2 (Figure 8A,C,D,F).

From all these data it results that SLC-0111 and AA-06-05 are able to impair tumor cell survival and migration, by activating the apoptosis pathway that passes through p-ERK1/2 and apoptosis marker activation. The antitumor activity of these compounds is corroborated by the shifting toward an epithelial phenotype of the cancer cells, and finally by the reduced ability to degrade the extracellular matrix, an event crucial for cell invasiveness.

## 3. Discussion

The pharmacological research is continuously working on defining new targets responsible for disease progression and characterizing novel effective drugs. In this scenario, CA-IX inhibition has been reported to be a potential target for novel anticancer therapeutic strategy. The role of CA-IX activity in several phases of cancer development (metabolic transformation, growth and progression, invasion and metastasis) implies the need to investigate CA-IX inhibition-related functional and biomolecular changes. 

Here we show an evident anti-proliferative and anti-invasive effect of SLC-0111 and AA-06-05 in A549 and MDA-MB-231 cancer cells. Through functional and molecular assays, we demonstrate the activation of apoptotic pathway and a decrease in invasion potential by CA-IX inhibition in cells maintained in normoxia.

It is known that CA-IX is expressed ectopically by almost all tumors, especially in the advanced stages [30]. The inhibition potential of two sulphonamide compounds, SLC-0111 and AA-06-05, has been initially evaluated in survival/proliferation experiments. A concentration-dependent effect on cell viability is observed only with AA-06-05 and with the highest doses of SLC-0111 (300 µM), as previously reported by Angeli and coworkers [19]. The reduced inhibitory effect of SLC-0111 on both tumor cells can be related with the more specific target of its inhibitory action, played mainly on CA-IX isoform, while the stronger inhibitory effect by AA-06-05 can be related to the inhibition of CA-I isozyme, beside the CA-IX isoform (Figure 1), because the CA-II has not been detected in our models. Indeed, it is well known the role of CA-I, II and IV isozymes in the regulation of cell respiration and acid/base homeostasis [8].

To strengthen this finding, the molecular mechanism controlling cell viability was evaluated. From a molecular point of view, the apoptotic pathway was investigated to find a molecular correlation between CA-IX inhibition and cell death, according with data on cell viability. Cancer cells use anaerobic glycolysis for energy intake, even in normoxic conditions, causing higher rates of glycolysis and increased production of CO_2_, H^+^, and lactate [31]. These metabolites have to be removed from the cells to prevent pHi acidification, and thus to maintain a slightly alkaline pHi consistent with survival conditions. Thus, considering the role of CA-IX in the regulation of cellular pH and, consequentially, ROS accumulation [21], we measured the expression of p-ERK1/2, a typical cell survival marker that in certain conditions behaves as an oxidative stress-induced apoptotic factor. Interestingly, an increased level of pERK1/2 is observed after 30 min of incubation with the high doses [100–200 µM] of the compounds in a manner independent on the concentration which is presumably the maximal effective.

Phosphorylated ERK1/2 allows the activation of the typical apoptotic factor, p53 [22,23,24], which expression is assessed in both tumor cell lines after 1 h of treatment with SLC-0111 and AA-06-05. According to the modulation of pERK1/2, an increase of p53 expression is overall evident, but no clear concentration dependent effect could be seen probably because the effect is already at the plateau. The trend of p53 expression shows a pERK1/2-indipendent modulation, because p53 may also be affected by other apoptotic factors, which remain to be investigated. If apoptotic process proceeds, Caspase 3 activation allows the definitive cell death. Indeed, this protein is activated by its cleavage and arrests the inhibition of particular transcription factors, whose action cause the degradation of nucleosome, last event of the apoptotic process [32]. A visible increase of cleaved Caspase 3 is evident in both tumor cells, after incubation for 4 h with AA-06-05, while in MDA-MB-231 the activation of Caspase 3 is visible also with the highest concentration of SLC-0111.

The overexpression of CA-IX is reported in many tumors to be correlated with EMT ability of cancer cells, but the mechanism of the invasion potential modulation, by CA-IX action, is only partially known. It has been reported that CA-IX action decreases the link between an adhesion protein, E-Cadherin, and the actin component of cytoskeleton, through the modulation of Rho-GTPase proteins, allowing the activation of ubiquitin degradation pathway of E-Cadherin [33]. The reduction of E-Cadherin determines the loss of cell-cell adhesion, allowing cell migration. This process, together with the acidification of pHe, which improves the action of metalloproteases on extracellular matrix (ECM) proteolysis, enhances the metastatic potential of cancer cells [30]. Both tumor cells used in this study are characterized by a high metastatic potential [34,35]. Therefore, to evaluate the action of SLC-0111 and AA-0605 on the invasion potential of MDA-MB-231 and A549 cells, we used the Boyden Chamber assay, a migration test in which cells treated with test compounds are exposed to a gradient of a chemotactic stimulus represented by serum. In basal condition we could observe a higher migration rate with MDA-MB-231, probably due to their mesenchymal phenotype. Following treatment of cell suspensions with both CA-IX inhibitors, an evident decrease of cellular migration is observed in MDA-MB-231 and A549 cells. In fact, in both MDA-MB-231 and A549 cells, a decrease of about 50% of cellular migration is observed with AA-06-05 compared to vehicle alone, while a decrease of about 25-30% of migration rate is reported in cells treated with SLC-0111. Overall, a higher efficacy of AA-06-05 is evident, according to the previous data collected through MTT assay, thus suggesting a combination effect between migration impairment and inhibition of cell survival/proliferation at overlapping concentration. These functional tests about CA-IX inhibition in cancer cells make evident that this enzyme is a valid target to develop new therapeutic anticancer approaches with a promising potential by these sulfonamide analogues. 

The overexpression of CA-IX in cancer cells, avoiding the link between E-Cadherin and the cytoskeleton, improves the exit of cells from original niches toward lymph and blood circulation. This process allows the invasion and the metastasis of cancer cells [36]. It is reported that an important event in the initiation of cancer metastasis is EMT, a process in which epithelial cells lose apical-basal polarity and gain a mesenchymal phenotype, through the increase of mesenchymal markers as Vimentin and Fibronectin, and decrease of epithelial ones, in this case E-Cadherin [26]. Moreover, the Boyden Chamber assay data on both tumor cell lines document a significant decrease of migration of cells treated with CA-IX pharmacological inhibitors. Thus, the expression of E-Cadherin and other proteins modulated during EMT process, such as Fibronectin and Vimentin, were assessed in both MDA-MB-231 and A549 cells. The incubation with SLC-0111 and AA-06-05 was performed in a longer timeline, 24 h, because the E-Cadherin metabolism implies the activation of ubiquitin-lysosomal degradation [37]. The mesenchymal phenotype of MDA-MB-231 has been confirmed by the lower basal expression of E-Cadherin, a main epithelial marker. Following treatment with CA-IX inhibitors, the increase of E-Cadherin expression is observed in both tumor cells, but mainly in MDA-MB-231 cells, where a clear concentration-dependent change is detected under treatment with both compounds. According to the increased E-Cadherin level, a decrease of Vimentin expression, the mesenchymal marker, is measured in both tumor cells lines, especially in MDA-MB-231. Fibronectin, a mesenchymal component of mammary tissues, is reported to cause the downregulation of E-Cadherin and the EMT process in breast cancer development [38,39]. However, in our experiments we did not observe a statistical significative reduction of mesenchymal marker Fibronectin. 

The normal activity of CA-IX allows the acidification of pHe, the lowest pHe causes activation and increases expression of proteinases and metalloproteases, like MMP-2 that degrade components of the ECM, facilitating invasion and migration [31,40,41]. Therefore, it is expected that the pharmacological inhibition of CA-IX would decrease MMPs activation and the invasiveness of cancer cells. To evaluate the modulation of MMPs activity, gelatin zymography was performed in the conditioned medium of both tumor cell lines treated with increased concentrations of SLC-01111 and AA-06-05. CA-IX inhibitors caused an evident decrease of MMP-2 activity, visualized as smaller and less intense degradation bands. These data are further confirmed by the release of TIMP-2, an endogenous inhibitor of the metalloprotease activity and tumor cell invasiveness [29,42,43]. In fact, an increase of TIMP-2 is observed under the same treatment condition that causes the decrease of MMP-2 activity, in both MDA-MB-231 and A549 cells. The decrease of MMP-2 activity and increase of TIMP-2 expression are more evident with the higher concentration of AA-06-05. Therefore, it is overall evident that the pharmacological inhibition of CA-IX through these sulfonamide compounds allows the decrease of invasive and metastatic potential of MDA-MB-231 and A549 cells.

Taken together, these data sustain that the pharmacological inhibition of CA-IX allows a meaningful increase of cell death and a decrease of aggressiveness and metastatic potential of tumor cells, verified also through the expression of apoptotic markers and the EMT pathway. Therefore, the results obtained may be a valid support to demonstrate the efficacy of future anticancer approaches based on the CA-IX inhibition obtained through SLC-0111 and AA-06-05 pharmacological action. Interestingly, the use of CA-IX inhibitors in vivo systems is not accompanied by non-specific toxicity [44]. This report, together with the advancements of clinical trials on some of these compounds (i.e., SLC-0111) [18], makes the field very interesting. The possible clinical perspectives are indeed twice: CA-IX overexpression would be a novel marker of cancer malignancy, and pharmacological inhibition of CA-IX may represent a novel anticancer therapy, alone or in association with other treatments. 

## 4. Materials and Methods 

### 4.1. Drugs and Reagents 

The pharmacological inhibitors of CA-IX tested were SLC-0111 and AA-06-05. The chemical properties and synthesis of the ureido benzene-sulphonamide, SLC-0111 (PubChem CID: 310360), have been described previously [17]. As already discussed, AA-06-05 is a SLC-0111 congener obtained by a divalent isosteric replacement approach, by means of introduction within the ureido moiety of a selenium element [19]. SLC-0111 has a good affinity for CA-IX, equal to a K_I_ of 45.0 nM calculated by stopped-flow carbon dioxide hydration assay, and poor affinity for CA-I and CA-II [17]. Inhibitor AA-06-05 has been developed from the lead compound SLC-0111, showing a K_I_ for CA-IX of 63.0 nM, for CA-I of 152.3 nM and for CA-II of 66.3 nM [19] (see structures in Figure 9).

Each compound was dissolved in dimethylsulphoxide (DMSO, Sigma-Aldrich St. Louis, MO USA); after the solubilization of the powders, sterile water was added to have a concentration of0.1 M. Starting from this stock solution, the working dilutions were prepared. All the solutions were prepared under sterile conditions and employing a vertical laminar flow safety hood. Once solubilized the stock solutions were stored at −20 °C. Cobalt chloride (CoCl_2_) was from Sigma (St. Louis, MO, USA). Anti-E-Cadherin (1:1000), anti-Vimentin (1:1000), anti-pERK1/2 (1:2000), anti-ERK1/2 (1:2000), anti-Cleaved Caspase-3 (1:1000) and anti-CA-II (1:500)were from Cell Signaling (Danvers, MA, USA). Anti-CA-IX (1:500) was from Merck KGaA (Darmstadt, Germany). Anti-p53 (1:500) and anti-TIMP-2 (1:2000) were from Santa Cruz Biotechnology (Dallas, TX, USA). Anti-Caspase-3 (1:1000), anti-Fibronectin (1:1000) and anti- β-actin (1:10,000) were from Sigma-Aldrich (St. Louis, MO, USA). Anti-CA-I (1:10,000) was from Abcam (Cambridge, UK).

### 4.2. Cell Cultures

Two tumor cells models were used: MDA-MB-231, metastatic breast cancer cells and A549, non-small cell lung cancer, both obtained from the American Type Culture Collection. Both tumor cell lines were grown in Dulbecco medium (DMEM 4500 High glucose, Euroclone, Milan, Italy) supplemented with 10% fetal bovine serum (FBS) (Euroclone, Milan, Italy) and 2 mM glutamine, 100 units/mL penicillin and 0.1 mg/mL streptomycin (Sigma Aldrich, St. Louis, MO, USA). The tumor cells were cultured in 10 cm diameter Petri dishes up to a confluent state, in a humidified normoxic atmosphere with 5% CO_2_. Cells were expanded through splitting 1:3 twice a week, for MDA-MB-231, and 1:4 twice a week for A549, both ones used until passage 30.

### 4.3. Tumor Cell Viability

The colorimetric quantitative assay of MTT was used to evaluate cell viability following treatment with CA-IX pharmacological inhibitors, and to analyze the potential cytotoxicity of the compounds [45,46]. A549 and MDA-MB-231 were seeded in 96-multiwell plates in medium with 10% FBS, at the density of 3 × 10^3^ cells in 100 µL for each well. After cell adhesion, cells were exposed to test compounds SLC-0111 and AA-06-05 at the concentrations of 10, 100, 200 and 300 µM in medium supplemented with 1% serum. Controls were medium with 0.1 and 1% FBS, and DMSO (the vehicle of compounds) corresponding at 300 µM of test compounds. Cell survival was assessed after 48 h. At the end of the stimulation, medium was removed and replaced with a solution of 1.2 mM MTT (Thiazolyl Blue Tetrazolium Bromide, Sigma-Aldrich St. Louis, MO, USA) in PBS (phosphate buffered saline, Euroclone, Milan, Italy), without phenol red, followed by an incubation of 4 h at 37 °C. At the end of incubation, the MTT solution was removed and replaced with DMSO to solubilize the formazan crystals. The intensity of the color was then detected through microplate absorbance reader (Infinite 200 Pro, Tecan Life Sciences, Switzerland) at 540 nm. Data were reported as 540 nm relative absorbance/well.

### 4.4. Tumor Cell Invasion

The Boyden Chamber procedure was used to evaluate cell invasiveness after treatment of the cells with different doses of CA-IX pharmacological inhibitors. The Neuro Probe 48-well micro-chemotaxis chamber (Nuclepore Corp., Pleasanton, CA, USA) was used [47]. The method is based on the passage of tumor cells across porous filters, with pores of 8 µm diameter (Neuro-Probe, Inc, Gaithersburg, MD, USA), from an upper compartment toward a lower compartment according a concentration gradient of the migration effector. Polyvinylpyrrolidone (PVP)-free polycarbonate filters, 8 µm pore size were coated with gelatin 1% (gelatin from bovine skin, type 1, Sigma Aldrich, St. Louis, MO, USA). Two different chemotactic solutions, composed by DMEM with 1% and 5% FBS, were placed in the lower wells. Then, 50 µL of cell suspension (2.5 × 10^4^ cells/mL) were added to each upper well. Before seeding, tumor cell suspensions were incubated with CA-IX inhibitors at 50 µM (or vehicle) at 37 °C for 30 min. Once assembled, the chamber was incubated at 37 °C for 18 h. Then, the filter was removed and fixed in methanol overnight. Cells were stained with Diff-Quik (BiomapSnc, Agrate B.za, MI, Italy), non-migrating cells on the upper surface of the filter were removed with a cotton swab, cut and mounted on glass coverslips. Migrated cells were counted using a light microscope (Nikon Eclipse E400 at 20× magnification) in 5 random fields per each well. Cell migration was measured by the number of cells moving across the filter. Each experimental point was done in triplicate and was calculated as mean value of migrated cells (±SD) for each experimental point.

### 4.5. Protein Expression

Protein expression was evaluated by western blot [48]. In total, 3 × 10^5^ cells were plated in 60 mm dishes, with complete DMEM supplemented with 10% FBS. After adhesion, subconfluent tumor cells were treated with 100 and 200 µM or 25–200 µM range of SLC-0111 and AA-06-05 in medium with 1% FBS or with CoCl_2_ (100 and 200 µM for 24 and 48 h). DMSO was tested at the corresponding higher concentration as a vehicle control. Time of incubation varied from 30 min to 12–24 h, according to the known kinetic activation of different proteins. At the end of each time, cells were washed with PBS twice, and lysed with 60 µL of CelLytic (Sigma-Aldrich St. Louis, MO USA) supplemented with Sodium orthovanadate (Na_3_VO_4_·2H_2_O) (1 mM) and a cocktail of protease inhibitors (Sigma Aldrich St. Louis, MO USA). After protein extraction and quantification, electrophoresis (50 μg of protein/sample) was carried out in 4–12% Bis-Tris Gels (Life Technologies, Carlsbad, CA, USA). Separated proteins were then blotted onto nitrocellulose membranes, incubated overnight with primary antibodies and then detected by enhanced chemiluminescence system (Biorad, Hercules, CA, USA). Results were normalized to those obtained by using an antibody against β-actin and quantified through Fiji ImageJ software. Data are analyzed as ratio of the arbitrary densitometry unit (A.D.U.) of target protein respect to the reference protein and, in most of the graphs, are reported vs. the basal control condition.

### 4.6. Gelatin Zymography

3 × 10^5^ cells/well were cultured in a 96 multiwell plates in DMEM supplemented with 10% FBS. After 18 h cells were exposed to increasing concentrations of SLC-0111 [50–200 µM] and AA-06-05 [25–200 µM] or DMSO as a negative control in serum free medium for 48 h at 37°C. Following centrifugation to remove cell debris, the conditioned medium was collected and stored at −20 °C. After protein quantification, samples (30 µL) were subjected to electrophoresis in 8% polyacrylamide gels and 1 mg/mL gelatin within non-reduction conditions, maintaining the running gels in a Running Buffer of TrisHCl 10× solution (Biorad, Hercules, CA, USA), at 4 °C. Gels were washed in a solution 2,5% Triton X-100 and incubated for 48 h at 37 °C in 50 mM Tris buffer containing 200 mM NaCl and 20 mM CaCl_2_, pH = 7.4. The gels were stained with 0.5% Coomassie brilliant blue R-250 (FlukaChemika and BioChemika, Deisenhofen, Germany) in 10% acetic acid and 45% methanol and 45% H_2_O MilliQ, de-stained with the same solution without Coomassie brilliant blue. Bands of gelatinase activity appeared as transparent areas against a blue background. Gelatinase activity (corresponding to MMP-2 with bands at 68–72 kDa) was evaluated by quantitative densitometry on inverted images normalizing each value for total proteins measured in each sample [49]. In parallel experiments TIMP-2 released in the conditioned medium was measured by western blot, correcting the optical density for the total proteins measured in each sample.

### 4.7. Statistical Analysis

The experimental data were obtained from the average of at least triplicate experiments and were reported with standard deviation (SD) of the experiments carried out. Student t test and ANOVA test were used to evaluate differences among groups. *p* < 0.05 was considered statistically significant.

## 5. Conclusions

CA-IX is strictly involved in regulation of acidic tumor microenvironment that is an emerging hallmark of cancer [3].

Our results, taken together, suggest a close relationship between CA-IX and cancer malignancy. Thus, CA-IX inhibition can be a valid target for the development of novel anticancer therapy, alone or in association with other treatments.

## Figures and Tables

**Figure 1 ijms-21-02983-f001:**
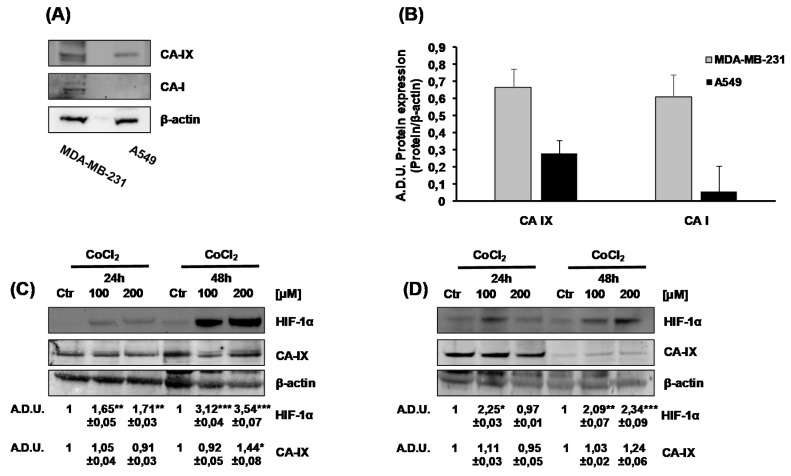
Carbonic anhydrase IX (CA-IX) and CA-I basal expression in MDA-MB-231 and A549 cells. (**A**). Protein expression of CA-IX and CA-I in MDA-MB-231 and A549 cells cultured for 24 h in complete medium with 1% FBS. (**B**). Quantification of CA-IX and CA-I protein expression. β-actin was used to normalize loading. Arbitrary Densitometry Units (A.D.U.) ± SD were reported as protein of interest vs β-actin. (*n* = 3). (**C**,**D**). HIF-1α and CA-IX protein expression in normoxic and hypoxic conditions in MDA-MB-231 (**C**) and A549 (**D**) cells. Tumor cells were treated with increasing doses of CoCl_2_ [100–200 µM], under experimental condition of DMEM with 1% FBS, for 24 and 48h. Numbers represent protein quantification reported as Arbitrary Densitometry Units (A.D.U.) ± SD of the protein of interest/β-actin vs the basal control condition (Ctr). (*n* = 3). * *p* < 0.05, ** *p* < 0.01 and *** *p* < 0.001 vs. untreated cells (Ctr).

**Figure 2 ijms-21-02983-f002:**
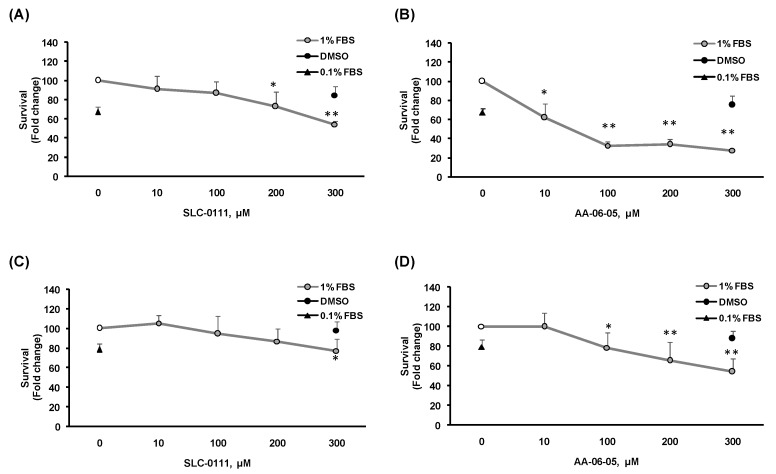
Survival curves of MDA-MB-231 and A549 cells exposed to CA-IX pharmacological inhibitors. MDA-MB-231 (**A**,**B**) and A549 (**C**,**D**) were treated with increasing concentrations [10–300 µM] of SLC-0111 and AA-06-05for 48 h, under experimental condition of medium with 1% FBS. Survival data were calculated as 540 nm relative absorbance/well. Data in the graphs are reported as fold change (means ± SD), giving 100% to the control condition of 1 % serum. (*n* = 3). * *p* < 0.05, ** *p* < 0.01 vs. untreated cells.

**Figure 3 ijms-21-02983-f003:**
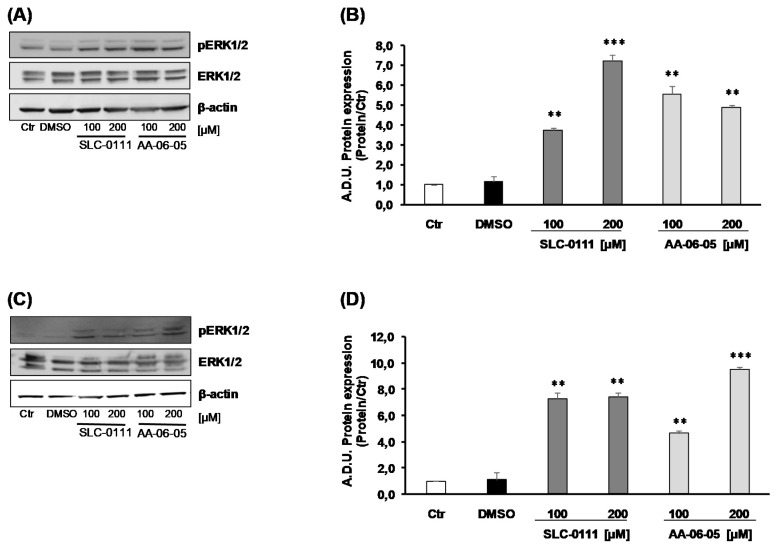
CA-IX pharmacological inhibitors increase the expression and activation of p-ERK1/2. MDA-MB-231 (**A**,**B**) and A549 (**C**,**D**) were treated with SLC-0111 and AA-06-05 [100–200 µM] for 30 min. The treatments were performed under experimental condition of medium with 1% FBS. Signals were evaluated through western blot and β-actin was used to normalize protein loading. For each experimental condition Arbitrary Densitometry Units (A.D.U.) ± SD were reported as pERK1/2/total ERK1/2 vs basal control. (*n* = 3). ** *p* < 0.01 and *** *p* < 0.001 vs. untreated cells (Ctr).

**Figure 4 ijms-21-02983-f004:**
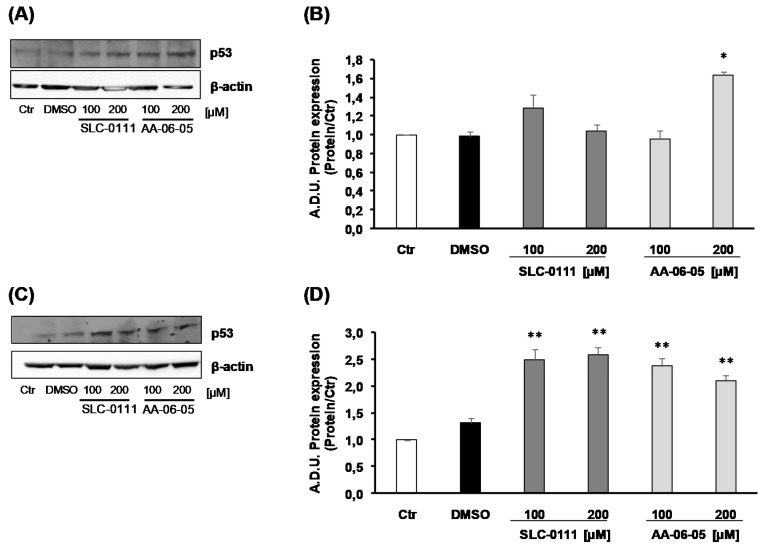
CA-IX inhibition allows the increased expression of apoptotic protein p53 in MDA-MB-231 and A549 cells. MDA-MB-231 (**A**,**B**) and A549 (**C**,**D**) were treated SLC-0111 and AA-06-05 [100–200 µM] for 1 h. The treatments were performed under experimental condition of medium with 1% FBS. Signals were evaluated through western blot and β-actin was used to normalize loading. For each experimental condition, Arbitrary Densitometry Units (A.D.U.) ± SD were reported as p53/β-actin vs basal control. (*n* = 3). * *p* < 0.05, ** *p* < 0.01 vs. untreated cells (Ctr).

**Figure 5 ijms-21-02983-f005:**
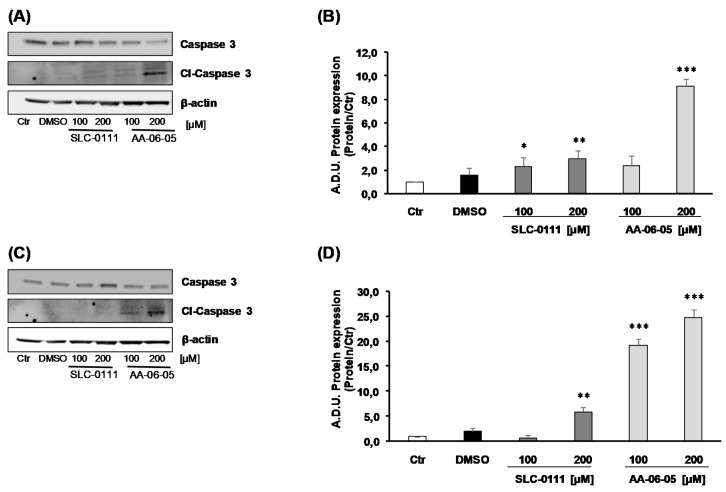
CA-IX inhibition allows the cleavage of Caspase 3 in MDA-MB-231 and A549 cells. MDA-MB-231 (**A**,**B**) and A549 (**C**,**D**) were treated SLC-0111 and AA-06-05 [100–200 µM] for 4h. The treatments were performed under experimental condition of medium with 1% FBS. Signals were evaluated through western blot and β-actin was used to normalize loading. For each experimental condition, Arbitrary Densitometry Units (A.D.U.) ± SD were reported as cleaved-Caspase-3/total Caspase 3 vs. basal control. (*n* = 3). * *p* < 0.05, ** *p* < 0.01 and *** *p* < 0.001 vs. untreated cells (Ctr).

**Figure 6 ijms-21-02983-f006:**
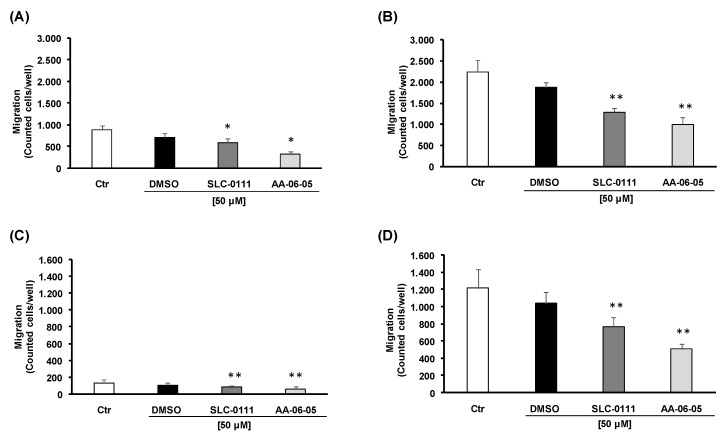
CA-IX inhibition reduces the migration ability of MDA-MB-231 and A549 cells. In suspension tumor cells were treated with a sub-toxic dose [50 µM] of SLC-0111 and AA-06-05, for 30min. To assess the migration ability of MDA-MB-231(**A**,**B**) and A549 (**C**,**D**), the Boyden Chamber assay was performed with cells suspended in basal condition of medium with 0.1% FBS, and toward two serum gradients: 1% FBS (**A**,**C**) and 5% FBS (**B**,**D**) added in the lower compartments. Data were reported as counted cells for each well (*n* = 3). * *p* < 0.05 and ** *p* < 0.01 vs. untreated cells (Ctr).

**Figure 7 ijms-21-02983-f007:**
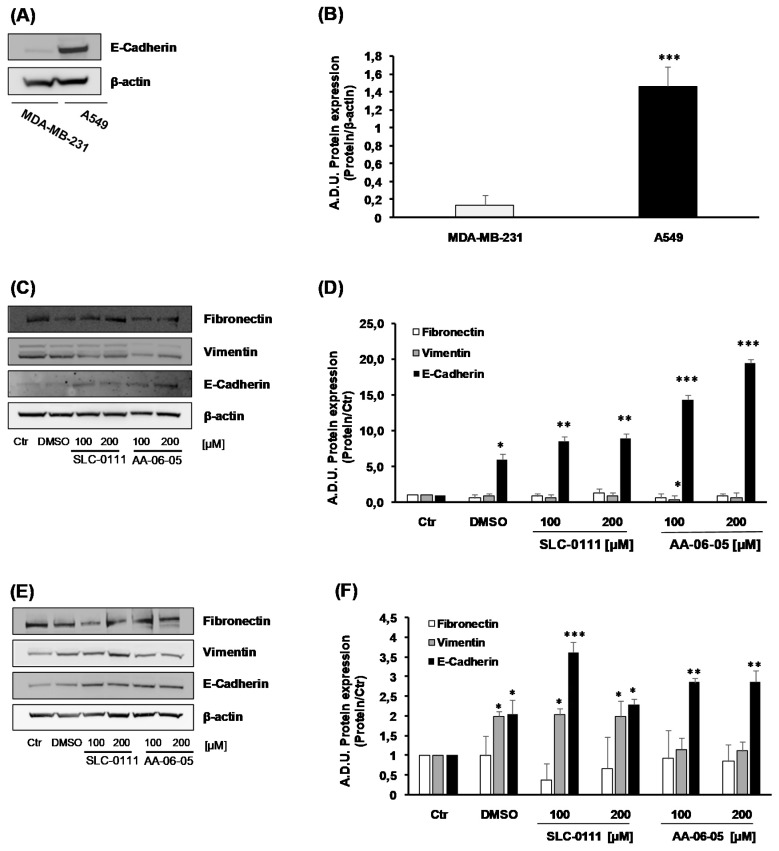
CA-IX pharmacological inhibitors control the mesenchymal phenotype of MDA-MB-231 and A549 cells. (**A**,**B**). Basal expression of E-Cadherin inMDA-MB-231 and A549 cells, cultured in medium with 1% FBS (*n* = 3). Arbitrary Densitometry Units (A.D.U.) ± SD were reported as protein of interest/β-actin. *** *p* < 0.001 A549 vs MDA-MB-231. (**C**,**D**). MDA-MB-231 and (**E**,**F**) A549 were treated with SLC-0111 and AA-06-05 [100–200 µM] for 24 h to evaluate the expression of cell-adhesion pathway: Fibronectin, E-Cadherin, Vimentin. The experiments were performed under condition of medium with 1% FBS. Signals were evaluated through western blot and β-actin was used to normalize protein loading. For each experimental condition Arbitrary Densitometry Units (A.D.U.) ± SD were reported as protein of interest/ β-actin vs. basal control. (*n* = 3). * *p* < 0.05, ** *p* < 0.01 and *** *p* < 0.001 vs. untreated cells (Ctr).

**Figure 8 ijms-21-02983-f008:**
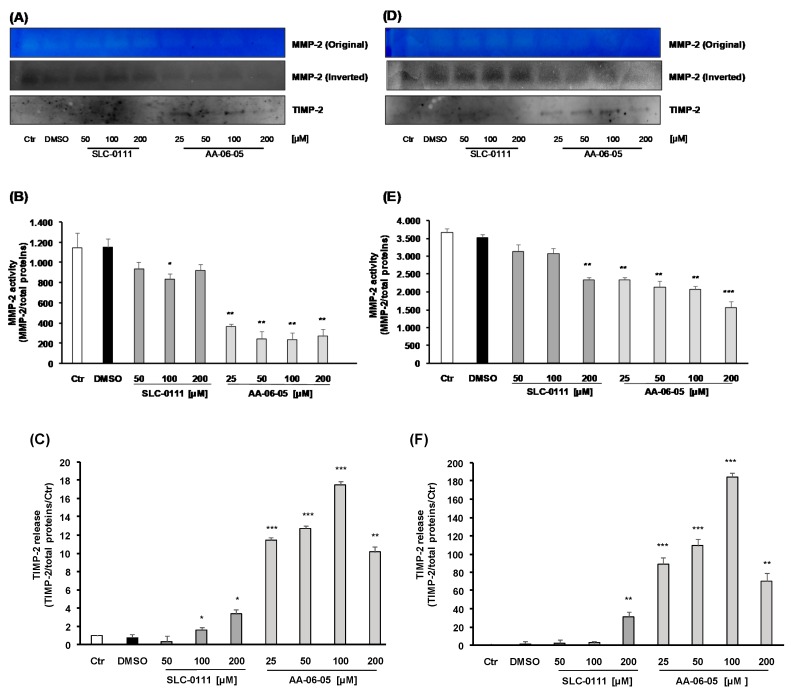
Zymographic analysis of conditioned medium from cultured MDA-MB-231 and A549 cells treated with CA-IX pharmacological inhibitors. Tumor cells were treated with increasing concentrations of SLC-0111 [50–200 µM] and AA-06-05 [25–200 µM], in serum-free medium. Conditioned medium was collected, in triplicate, after 48h and analyzed by gelatin zymography. MMP-2 activity in conditioned media of MDA-MB-231 (**A**,**B**) and A549 (**D**,**E**) was assessed as lysis bands on blue background and quantified in the inverted images as optical density (± SD)/total proteins in the sample. (*n* = 3). * *p* < 0.05, ** *p* < 0.01 and *** *p* < 0.001 vs. untreated cells (Ctr). Western blot analysis was performed in parallel experiments to evaluate the release of TIMP-2 in the conditioned medium of MDA-MB-231 (**C**) and A549 (**F**). For each experimental condition TIMP-2 release was reported as optical density (± SD)/total proteins in the sample vs. basal control. (*n* = 3). * *p* < 0.05, ** *p* < 0.01 and *** *p* < 0.001 vs. untreated cells (Ctr).

**Figure 9 ijms-21-02983-f009:**
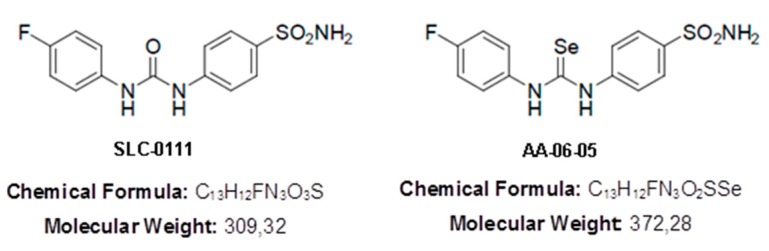
SLC-0111 and AA-06-05 chemical structures.

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
