# Peer review of "Pharmacological Inhibition of CA-IX Impairs Tumor Cell Proliferation, Migration and Invasiveness"

_ijms, 2020, doi:10.3390/ijms21082983_

Round 1

Reviewer 1 Report

In the submitted manuscript:  “Pharmacological inhibition of CA-IX impairs tumor cell proliferation, migration and invasiveness“  Valerio Ciccone et al. claim that: “Collectively, these data confirm the potential use of CA-IX 25 inhibitors, and in particular SLC-0111 and AA-06-05, as agents to be further developed, alone or in combination with other conventional anticancer agents.”

However, the experimental data presented in the figures do support these claims rather marginally:

  1. In Fig. 1A the CA IX band in MDA-MB-231cells is a double band, in A549 cells is a single band. In case that in MDA-MB-231cells part the CA IX is phosphorylated, in A549 cells is not phosphorylated?
  2. In the same Fig. 1A the CA I positivity is a triple band (one major, two minor bands). This raises the question about the Ab specificity
  3. In the Fig. 1C after incubation for 48 hours in the presence 100 µM CoCl2 there is a drop in the intensity of the CA IX band in comparison to the control. However, the authors claim that: “we did not observe an activation of CA-IX 83 expression in both cell lines”. Of course this is not an activation it is rather a “deactivation” of the CA IX protein expression.  Moreover, one can see (sometimes) double bands. However, the intensity of the bands is exchanged in contrast to the Fig 1A.
  4. In the Fig. 1C the line with HIF-1α is shifted to left. Is this the same gel?
  5. In Fig. 3B there is not a difference (even a small drop) between 100 and 200µM AA-06-05, the same is true in Fig. 3D for 100 and 200 µM SLC-0111. In Fig. 3C ERK1/2 started to be a “triple” band. In other words, this process seems to be 100 and 200µM independent.
  6. In Fig. 4B again there is not a difference between 100 and 200µM SLC-0111, in Fig. 4D there is not a difference between 100 and 200µM both SLC-0111 and AA-06-05. (In AA-06-05 there is even a small drop). In other words, this process seems to be 100 and 200µM independent.
  7. In Fig. 5A the “addition” of 50µM AA-06-05 track is confusing the reader and is not acceptable. (It is a “single” addition in the whole Figs).
  8. In Fig. 5D the effect of 100 µM SLC-0111 is below(?!) both controls.
  9. The quality of the Fig. 8A and Fig. 8D is simply not acceptable for this kind of publication. One has to enlarge them up to 800% to see/not to see the results.

Based on the facts mentioned above I do not recommend to accept this manuscript for publication.  The idea presented in the paper is interesting, nevertheless, this kind of the results presentation is not convincing and supports it only marginally.

Author Response

In the submitted manuscript:  “Pharmacological inhibition of CA-IX impairs tumor cell proliferation, migration and invasiveness“  Valerio Ciccone et al. claim that: “Collectively, these data confirm the potential use of CA-IX 25 inhibitors, and in particular SLC-0111 and AA-06-05, as agents to be further developed, alone or in combination with other conventional anticancer agents.”

However, the experimental data presented in the figures do support these claims rather marginally:

  • In Fig. 1A the CA IX band in MDA-MB-231cells is a double band, in A549 cells is a single band. In case that in MDA-MB-231cells part the CA IX is phosphorylated, in A549 cells is not phosphorylated?
  1. We thank the referee for the accurate revision of our data and evidence of misleading points. As reported on the datasheet of the antibody used(Datasheet: anti-CAIX,MABN713, Merck KGaA, Darmstadt, Germany) “uncharacterized bands may appear in some lysates.” Therefore, the different number of bands may depend on the different cell lines analyzed: in MDA-MB-231 (with more intense expression) there is a doublet, while in A549 there is a single band. Moreover, the antibody recognizes total CA-IX and does not select a determinate post-translational modification.
  1. In the same Fig. 1A the CA I positivity is a triple band (one major, two minor bands). This raises the question about the Ab specificity.
  1. We appreciate the tip of the reviewer, but the antibody used to evaluate CA-I expression (ab108367, Abcam, Cambridge, UK) was chosen considering both the literature and the commercial availability. At molecular weight of CA-I (29 kDa), indicated in the datasheet,no signal is evident inA549 but is clearly evident in MDA-MB-231, with the single net band. As above, different lysates can generate uncharacterized bands. Therefore, we conclude that A549 have no expression of CA-I enzyme.
  1. In the Fig. 1C after incubation for 48 hours in the presence 100 µM CoCl2 there is a drop in the intensity of the CA IX band in comparison to the control. However, the authors claim that: “we did not observe an activation of CA-IX expression in both cell lines”. Of course this is not an activation it is rather a “deactivation” of the CA IX protein expression. Moreover, one can see (sometimes) double bands. However, the intensity of the bands is exchanged in contrast to the Fig 1A.
  1. Thank you for careful text analysis. Indeed, the word “activation” was inappropriate and has been conveniently replaced in the manuscript with “increase” (Line 82). The different intensity of the bands depends on the biological variability in lysates of different experiments involving different passages of the same cells.

(Fig 1C) The drop visible in CA-IX expression seems to be a lower expression of the target, but this decreaseis attributable toa small difference in sample loading. Normalizing the blot quantification on its own β-actin (the housekeeping protein), the difference proves to be minimal and not significant. We now have added the density quantification into the figure.

  1. In the Fig. 1C the line with HIF-1α is shifted to left. Is this the same gel?
  1. We lined up the two different gels in which the same samples were analyzed.
  1. In Fig. 3B there is not a difference (even a small drop) between 100 and 200µM AA-06-05, the same is true in Fig. 3D for 100 and 200 µM SLC-0111. In Fig. 3C ERK1/2 started to be a “triple” band. In other words, this process seems to be 100 and 200µM independent.
  1. We appreciate the comment of the reviewer and we apologize for the inaccuracy. We modified this part of discussion in the manuscript, specifying that there is no concentration dependent effect by our molecules on ERK1/2 phosphorylation (Lines 283-284).

Different number of visible bands of these signals depends on the different lysate, as reported in Cell Signaling references about the antibodies used,anti-pERK1/2 (91065, Cell Signaling Danvers, MA USA) and anti total-ERK1/2 (91025, Cell Signaling Danvers, MA USA), (https://www.ncbi.nlm.nih.gov/pmc/articles/PMC3392104/figure/F1/) and literature. In MDA-MB-231 the signal appears as a clear double band, while in A549 appears as a triple band.

  1. In Fig. 4B again there is not a difference between 100 and 200µM SLC-0111, in Fig. 4D there is not a difference between 100 and 200µM both SLC-0111 and AA-06-05. (In AA-06-05 there is even a small drop). In other words, this process seems to be 100 and 200µM independent.
  1. Based on the comment we modified the manuscript to clarify the p53 modulation, according to the graphic data, documenting that there is no concentration dependent effect (Lines 287-289).

In Fig. 5A the “addition” of 50µM AA-06-05 track is confusing the reader and is not acceptable. (It is a “single” addition in the whole Figs).

  1. We agree with the reviewer and, to avoid reader confusion, the 50 uM data in this figure panel have been delated.
  1. In Fig. 5D the effect of 100 µM SLC-0111 is below(?!) both controls.
  1. We agree with the observation of the reviewer, the result reports the normalization of the optical density obtained by Image.j software, considering the dishomogeneity of the background. The lower value at 100µM of SLC-0111 is however not statistically significant respect to the two controls.
  1. The quality of the Fig. 8A and Fig. 8D is simply not acceptable for this kind of publication. One has to enlarge them up to 800% to see/not to see the results.
  1. We apologize for the barely visible signals, what the reviser sees for MMP-2 proteolytic activity are the same original images, transformed into black and white, to facilitate a greater visibility of the signal. In order to improve the visibility, we enhancedthe quality and enlarged the images, as possible, enclosing also the original images of the zymography in figure 8. Moreover, we know that the expression of TIMP, endogenous inhibitor of the metalloproteases released in the medium, is a faint signal. Thus, what images fail to make evident is highlighted by the graphs, calculated as optical density by Fiji. ImageJ software.

Based on the facts mentioned above I do not recommend to accept this manuscript for publication.  The idea presented in the paper is interesting, nevertheless, this kind of the results presentation is not convincing and supports it only marginally.

  1. We believe that the quantity of data (produced on two different cell lines representing two types of solid tumors) and the variety of functional and molecular assays performed, going into detail on the mechanisms underlying the functional responses, cannot considered marginal. Our data moreover strength the clinical data obtained with SLC-0111, which demonstrates its safety in Phase I clinical trial (reference n. 18) [A Phase 1 Study of SLC-0111, a Novel Inhibitor of Carbonic Anhydrase IX,in Patients With Advanced Solid Tumors. McDonald PC et. al.,Am J Clin Oncol. 2020 Apr 2.] and is now under study for antitumor efficacy in phase Ib clinical trial together with a conventional cytotoxic drug [A Study of SLC-0111 and Gemcitabine for Metastatic Pancreatic Ductal Cancer in Subjects Positive for CAIX (Identifier: NCT03450018)].

Reviewer 2 Report

Introduction. "We took advantage of the availability of novel CA IX inhibitors, SLC-0111 and AA-06-05.  SLC-0111 is a ureido benzene-sulfonamide, currently in phase 2 clinical trial for the treatment of solid tumors associated to hypoxic micro-environments [16,17]. SLC-0111 has a good affinity for this enzyme, equal to a KI of 45.0 nM calculated by stopped-flow carbon dioxide hydration assay.  Inhibitor AA-06-05 has been developed from the lead compound SLC-0111 by its bioisosteric modification, i.e., the replacement of the oxygen atom from the ureido functionality by a selenium one, showing a KI for CA-IX of 63.0 nM, for CA-I of 152.3 nM and for the CA-II of 66.3 nM [18] . "

This final part of introduction is more suitable for Material and Methods...

On the other side, since authors examined the basal expression of CA-I, CA-II and CA-IX in their results, some information on the role of CA-I and  II isozymes would be useful.

Discussion. "In fact, in both MDA-MB-231 and A549 cells, a decrease of about 50%  of cellular migration is observed with AA-06-05 compared to vehicle alone, while a decrease of about  25-30% of migration rate is reported in cells treated with SLC-0111. This different behavior can come  from the different nature of the two cancer cell lines, indeed MDA-MB231 have mesenchymal origin."

This part of discussion describes similar effects of tested compouds on BOTH CELL LINES and subsequently their "different behavior" is explained  by the different nature of the two cancer cell lines.

"we measured the expression of the oxidative stress-induced apoptotic factor p-ERK1/2."

ERK1/2 generally promotes cell survival; but under certain conditions, ERK1/2 can have pro-apoptotic functions. Please, don't present it as univocally apoptotic factor...

Results."Data showed that the  expression of p53 increased with SLC-0111 and AA-06-05, compared to controls in both  MDA-MB-231 (Figure 4A and B) and A549 (Figure 4C and D)."

According to the figure 4, , p53 expression increased (statistically significantly) only with 200nM concentration of AA-06-05  and not with SLC-0111 in MDA-MB-231 cells. Please correct the text accordingly.

Questions:

1.What is the affinity of SLC-0111 to CA-II and CA-I?

2.Using 2 controls (0.1% FBS and DMSO), how was cell survival calculated?

3. From Fig. 2B it seems that DMSO had quite significant effect on cell viability... please comment.

4. Can the effect of AA-06-05 on migration of MDA-MB-231 cells be caused by its inhibition of proliferation at 50uM concentration (according to MTT results)? Please exclude this possibility or show the same effect with lower concentration, not affecting cell proliferation.

Author Response

Introduction. "We took advantage of the availability of novel CA IX inhibitors, SLC-0111 and AA-06-05.  SLC-0111 is a ureido benzene-sulfonamide, currently in phase 2 clinical trial for the treatment of solid tumors associated to hypoxic micro-environments [16,17]. SLC-0111 has a good affinity for this enzyme, equal to a KI of 45.0 nM calculated by stopped-flow carbon dioxide hydration assay. Inhibitor AA-06-05 has been developed from the lead compound SLC-0111 by its bioisosteric modification, i.e., the replacement of the oxygen atom from the ureido functionality by a selenium one, showing a KI for CA-IX of 63.0 nM, for CA-I of 152.3 nM and for the CA-II of 66.3 nM [18] . "

This final part of introduction is more suitable for Material and Methods...

  1. We appreciated the useful suggestion. Now we have included this information in “Materials and methods” (Lines 369-374).

On the other side, since authors examined the basal expression of CA-I, CA-II and CA-IX in their results, some information on the role of CA-I and II isozymes would be useful.

  1. Thank you for your tip, we added more details about these two isoenzymes (Lines 66-69), previously not deepened because they are not the main target of our study.

Discussion. "In fact, in both MDA-MB-231 and A549 cells, a decrease of about 50%  of cellular migration is observed with AA-06-05 compared to vehicle alone, while a decrease of about 25-30% of migration rate is reported in cells treated with SLC-0111. This different behavior can come  from the different nature of the two cancer cell lines, indeed MDA-MB231 have mesenchymal origin."

This part of discussion describes similar effects of tested compouds on BOTH CELL LINES and subsequently their "different behavior" is explained by the different nature of the two cancer cell lines.

  1. We agree with this comment. The difference between the two cell lines resides in the greater migration capacity of MDA-MB-231 respect to A549, explained by the mesenchymal nature of MDA-MB-231. The decrease of migration ability, induced by the inhibitors, follows the same trend in both tumor cells. We have modified the manuscript (Discussion lines 307-308) to clarify it.

"we measured the expression of the oxidative stress-induced apoptotic factor p-ERK1/2."

ERK1/2 generally promotes cell survival; but under certain conditions, ERK1/2 can have pro-apoptotic functions. Please, don't present it as univocally apoptotic factor...

  1. Based on the reviewer comment, we have added further specifications in the text regarding the role of pERK both in the results (Lines 125-126) and in the discussion (Lines 280-281).

Results."Data showed that the expression of p53 increased with SLC-0111 and AA-06-05, compared to controls in both  MDA-MB-231 (Figure 4A and B) and A549 (Figure 4C and D)."

According to the figure 4, p53 expression increased (statistically significantly) only with 200nM concentration of AA-06-05  and not with SLC-0111 in MDA-MB-231 cells. Please correct the text accordingly.

  1. We appreciate the tip of the reviewer and the modulation of p53 has been clarified in the results of the manuscript (Lines 142-143).

Questions:

1.What is the affinity of SLC-0111 to CA-II and CA-I?

  1. According to Pacchiano et al., J Med Chem. 2011 Mar 24;54(6):1896-902.(ref. n. 17) ,the affinity of SCL-0111 for the other two isoforms is Ki (CA-I) = 5080 nM and Ki (CA-II) = 960 nM, thus not relevant for CA-I and CA-II enzyme inhibition.

2.Using 2 controls (0.1% FBS and DMSO), how was cell survival calculated?

  1. From Fig. 2B it seems that DMSO had quite significant effect on cell viability... please comment.
  2. We appreciate the comment. We clarified in the text how the absorbance data were processed (Lines 113-115). All cell viability data were normalized and analyzed using untreated cells (medium with 1% FBS) as control. The values related to DMSO, as evident from the diagrams, scarcely and not significantly differ from that of basal control, assuming no or little influence of DMSO on cell viability especially at the higher concentrations of the test substances. In figure 2B, in particular, the level of cell viability decrease is 75% under 300 µM AA-06-05 vs 27% under the corresponding DMSO concentration.
  3. Can the effect of AA-06-05 on migration of MDA-MB-231 cells be caused by its inhibition of proliferation at 50uM concentration (according to MTT results)? Please exclude this possibility or show the same effect with lower concentration, not affecting cell proliferation.
  4. We cannot exclude that the two functional parameters of MDA-MB-231 are both affected by 50 uM of AA-06-05. Indeed, in the lines 312-315, we already suggested a “combination effect” between migration and proliferation/survival modulation, even ifthe cells were exposed to the inhibitors in different conditions. In fact, they were incubated with sub toxic dose of AA-06-05 for 18h in migration experiments, while for 48h in MTT assays, in which the compound shows the major cytotoxic effect at lower concentrations. In vivo proliferation and migration/invasiveness cannot be split, and we do not think that doing further experiments (at present not possible due to COVID-19 institutional constraints) will add clue information to this topic.

Round 2

Reviewer 1 Report

Please can can you explain this sentence:

In basal condition we could observe a 307 higher migration rate with MDA-MB-231, probably due to their mesenchymal origin

(MDA-MB-231 express Vimentin, but lacks N cadherin, which is a typical mesenchymal marker.)  

A number of breast cancer cell lines that can be classified as mesenchymal-like.  However, they  are epithelial in origin.

Author Response

Please can can you explain this sentence:

In basal condition we could observe a 307 higher migration rate with MDA-MB-231, probably due to their mesenchymal origin

(MDA-MB-231 express Vimentin, but lacks N cadherin, which is a typical mesenchymal marker.)  

A number of breast cancer cell lines that can be classified as mesenchymal-like.  However, they  are epithelial in origin.

Thank you for the comment. In the text the word “origin” has been changed in “phenotype” (Line 308). As reported on ATCC® website, MDA-MD-231 have an epithelial origin, but they derive from a metastatic site in which they acquired the mesenchymal phenotype. The transition from epithelial to mesenchymal state is described by Tam &Weinberg in “The epigenetics of epithelial-mesenchymal plasticity in cancer” (https://doi.org/10.1038/nm.3336) and by Gupta & Massagué in “Cancer metastasis: building a framework” (https://doi.org/10.1016/j.cell.2006.11.001). Therefore, MDA-MB-231 is an in vitro model of breast cancer recapitulating a mesenchymal phenotype. This is supported by the lower basal expression of E-Cadherin, compared to A549, and higher rate of migration in the Boyden Chamber assay.